# Graviception Uncertainty, Spatial Anxiety, and Derealization in Patients with Persistent Postural-Perceptual Dizziness

**DOI:** 10.3390/jcm13226665

**Published:** 2024-11-06

**Authors:** Kathrine Jáuregui-Renaud, Rodrigo Cabrera-Pereyra, José Adán Miguel-Puga, Mónica Alcántara-Thome

**Affiliations:** 1Unidad de Investigación Médica en Otoneurología, Instituto Mexicano del Seguro Social, Ciudad de México 06720, Mexico; rodrigo.cabrerap@gmail.com (R.C.-P.); adan.miguel@imss.gob.mx (J.A.M.-P.); 2Hospital de Especialidades del Centro Médico Nacional Siglo XXI, Instituto Mexicano del Seguro Social, Ciudad de México 06720, Mexico; dramonicaalcantara@gmail.com

**Keywords:** Persistent Postural-Perceptual Dizziness, otoliths, postural control, spatial anxiety, derealization

## Abstract

**Objectives**: Persistent Postural-Perceptual Dizziness (PPPD) is a frequent diagnosis in patients with chronic dizziness, ineffective postural control, visual dependence, and emotional symptoms. **Methods**: 53 patients with PPPD (25–84 years old) and 53 adults (29–84 years old) with no vestibular disease agreed to participate in this study. Assessments included: vestibular function tests (sinusoidal yaw rotation and vestibular-evoked myogenic potentials); accuracy and precision of Subjective Visual Vertical (SVV) estimation while static and during on-axis yaw rotation; static posturography with open/closed eyes and 30° neck extension, while standing on hard/soft surface; questionnaires on symptoms of unsteadiness, spatial anxiety, dizziness-related handicap, anxiety/depression, depersonalization/derealization, and perceived stress. After preliminary bivariate analyses, analysis of covariance was performed on the measurements of postural sway, spatial anxiety, and dizziness-related handicap (*p* < 0.05). **Results**: Higher intraindividual variability (reduced precision) on SVV estimations was evident in patients with PPPD compared to adults with no vestibular disease, which was related to the length of postural sway, to velocity displacement in the sagittal plane, as well as to spatial anxiety and common mental symptoms (including depersonalization/derealization symptoms). Covariance analysis showed contribution of these factors to the dizziness-related handicap reported by the patients. **Conclusions**: Unprecise graviception could be a contributing factor to the postural instability and mental symptoms reported by patients with PPPD, which in turn contribute to their dizziness-related handicap.

## 1. Introduction

In patients with chronic dizziness, Persistent Postural-Perceptual Dizziness (PPPD) may be the most frequent diagnosis [1,2,3]. The term comprises a variety of syndromes that occur long after acute vestibular symptoms or disruption of balance, and are related to ineffective postural control strategies, with visual dependence and emotional symptoms [4,5,6,7]. Imaging studies support that PPPD is related to altered regional cerebral blood flow in the insular, frontal, and cerebellar cortices [8], and reduced connectivity among cortical areas that are involved in vestibular processing and in spatial cognition [9,10]. Recent concepts incorporate predictive processing of sensory inputs and prioritization of postural stability to explain the symptoms (for review [11]). The clinical manifestations include “one or more symptoms of dizziness, unsteadiness, or non-spinning vertigo that are present on most days for three months or more and are exacerbated by upright posture, active or passive movement, and exposure to moving or complex visual stimuli” [12].

Sway recordings in static upright posture have shown that patients with PPPD may have visual dependency to maintain balance [13]. Balance control on Earth requires both vestibular and somatic signals to provide the perception of the head–body posture relative to gravity, while continuous multisensory combination allows the transformation of visual information from an eye-centered reference frame into gravity-centered reference [14] and assists the motor activity needed to compensate the force of gravity [15,16]. Bayesian framework suggests that the central nervous system constructs estimates of the sensorimotor transformations to optimally respond to environmental stimuli, and to represent the structure of the uncertainty in the inputs, outputs, and in the transformations themselves [17]. During upright stance, when the standing surface is unreliable as a reference, the vestibular system orients the upper body with respect to gravity particularly at velocities around normal postural sway [16]. Among other parameters, the velocity displacement of the center of pressure in the sagittal plane (VFY) allows monitoring of the short length–high velocity compensating movements used to maintain the upright position [18].

Patients with PPPD can have dysfunction of otolith information within compound stimuli [19,20]. The subjective visual vertical (SVV) is a psychophysical measure of the deviation of an individual perception of vertical from the gravitational vertical [17,21] with little proprioception contribution in the static upright position [22]. Conditions to evaluate SVV are varied, including measurements in static conditions (either upright or with head tilt), or during motion (on-axis or off-axis yaw rotation) [21]. However, SVV accuracy can be modified by visual feedback and cognitive processes, while SVV precision depends mostly on otolith input, and it is not affected by visual feedback [23]; accuracy can be assessed by the mean deviation error while precision can be assessed by the variability of intraindividual estimations [24]. In a static upright position in darkness, when other sensory inputs are minimized, the precision of SVV is limited by the effectiveness of the otolith organs and by central computational mechanisms [17].

Gravity estimations are also essential for spatial orientation and navigation. Spatial representations are encoded and processed to subserve motor activity in the environment (for review [25]), entailing a variety of cognitive functions, such as mental rotation, spatial imagery, memory, attention, and navigation (for review [26]). Space anxiety is the domain-specific anxiety that is related to the fear and apprehension felt when performing spatial tasks [27]. Patients with bilateral vestibular dysfunction may report spatial anxiety related to navigation [28], while spatial anxiety and perspective-taking may contribute to the dizziness-related handicap reported by patients with a variety of peripheral vestibular disorders [29]. Though patients with PPPD may display difficulties in constructing and/or using cognitive maps [30], a review of the literature showed no studies on spatial anxiety in patients with PPPD.

In this study we aimed to assess the correlation between the accuracy and precision of SVV with (1) the postural sway in static upright-position in varied conditions, with (2) spatial anxiety (three domains), and (3) their impact on the dizziness-related handicap reported by adults with PPPD, compared to adults with no vestibular disease, including the following cofactors: age, sex, symptoms of common mental disorders (anxiety, depression, depersonalization/derealization), and perceived stress.

## 2. Materials and Methods

### 2.1. Participants

In a neuro-otology clinic, after approval by the institutional Research and Ethics Committees (2023-3601-025), 106 consecutive participants fulfilling the selection criteria (explained below) gave their written informed consent to participate in this study. There were 53 patients diagnosed with PPPD (25 to 84 years old; 40 women and 13 men), and 53 volunteers (family/companions) with no history or clinical evidence of vestibular disease (29 to 84 years old, 40 women and 13 men). No evidence of vestibular dysfunction was observed in participants without PPPD on both their clinical records and the vestibular evaluations performed to participate in the study. All the participants have completed at least nine years of formal school (secondary school) and were naive to the study protocol. The sample size was calculated to assess a R^2^ of 0.1 for the dependent variable, and R^2^ of 0.3 for cofactors, with bilateral type I error of 0.05 and type II error of 0.2 [31].

PPPD was diagnosed for the first time and all the patients fulfilled the diagnostic criteria outlined by the Committee for the Classification of Vestibular Disorders of the Bárány Society [12]. All reported their last episode of spinning-vertigo at least four months before participating in the study, among them 10 patients had a previous diagnosis of unilateral vestibular dysfunction (acute vestibulopathy) and 20 of Benign Paroxysmal Positional Vertigo (BPPV), while the remaining 23 had no specific vestibular diagnosis but reported a previous history of dizziness/vertigo, none of them had history of Meniere’s Disease. All the participants (with or without PPPD) denied having a history or medical record of middle ear/neurological (including migraine)/retinal/autoimmune or autonomic disorders, hearing loss >25 dBHL, or submission to psychiatric care or psychopharmacological treatment.

The general characteristics of the participants are described in Table 1. The report of tobacco or alcohol use was low (with no alcohol abuse), but one patient reported consuming 11 cigarettes per day. In the two groups, the report of corrected refraction errors was similarly high, while the frequency of systemic high blood pressure was circa 20%, and the frequency of diabetes was <10%; however, in patients with PPPD, diabetes mellitus was always comorbid with systemic high blood pressure.

### 2.2. Procedures

After assessments on general health and personal habits (alcohol and tobacco use) by an in-house questionnaire and direct interview, the following questionnaires were administered for self-report:A standardized questionnaire of symptoms related to unsteadiness [32] that includes nine items with no/yes responses; 0 points are scored for each ‘‘no’’ response and one point is scored for each ‘‘yes’’ response, except for vertigo which is scored two points. Frequent falls are considered only when reported ≥1 per month, and frequent stumbles are considered only when reported ≥1 per week. A total score is calculated by summing the scores for the nine items (range 0 to 10). Scores ≥ 4 points are related to balance disorders [32].The Hospital Anxiety and Depression Scale (HADS) [33] that comprises 14 items, 7 for anxiety and 7 for depression; each item is rated on a 4-point scale (0 to 3), and each subscore ranges from 0 to 21, while a total score is obtained by summing the ratings for all the items (range 0 to 42). A cut-off score of ≥8 for the two subscores and ≥11 for the total score have shown sensitivities and specificities ≥ 0.70, and Cronbach’s alpha coefficient ≥0.67 [34].The Depersonalization/Derealization Inventory [35] comprises 28 items rated on a scale from 0 to 4. The total score is calculated by summing all the individual scores (range 0 to 112), with a Cronbach’s alpha coefficient of 0.95. [35].The Perceived Stress Scale-10 (PSS-10) [36] contains 10 items that are coded on a 5-point scale (0 to 4). A total score is computed by reverse scoring the four positively worded items and then summing all the scale items (range 0 to 40), with a Cronbach’s alpha coefficient of 0.78 [37].The Spatial Anxiety Scale [38] comprises 24 items, including three subscales to assess anxiety about spatial imagery (8 items), spatial navigation (8 items), and spatial mental manipulation (8 items), that are related to objective performance in the relevant spatial subdomain. Items are scored on a 4-point scale (0 to 4). The scale total score and subscores are obtained by summing the ratings for all the scale items and the ratings for each subscale, respectively, with Cronbach’s alpha coefficient >0.8 [38].The Dizziness Handicap Inventory [39] comprises 25 items to assess the self-perceived handicapping effects imposed by dizziness and unsteadiness. The items are sub-grouped into three content domains representing functional (9 items), physical (7 items), and emotional (9 items) aspects of handicap, which are scored on a 4-point scale (0 to 4). A total score is obtained by summing the ratings for all the items (range 0 to 100) [39], with Cronbach’s alpha coefficient ≥0.88 [40].

After completing the questionnaires, vestibular evaluation was performed during the morning or early afternoon, including:Static posturography (Posturolab 40/16, Medicapteurs, Balma, France) was carried out using eight different sensory conditions: 1. hard surface/open eyes (1st baseline), 2. hard surface/closed eyes; 3. hard surface/open eyes (2nd baseline), 4. hard surface/open eyes/30° neck extension; 5. soft surface/open eyes (3rd baseline), 6. soft surface/closed eyes; 7. soft surface/open eyes (4th baseline), 8. soft surface/open eyes/30° neck extension. Estimations of the 90% confidence ellipse of the sway area, the length of sway, and the velocity displacement of the center of pressure in the sagittal plane (VFY), which monitors short length–high velocity compensating movements [18], were obtained by the software provided by the manufacturer of the platform (Medicapteurs, Balma, France).Sinusoidal rotation at 0.16 Hz and at 1.28 Hz (60°/s peak velocity), and SVV estimation in the upright sitting position during two conditions, static and on-axis yaw rotation (300°/s) (I-Portal-NOCT-Professional, Neuro-Kinetics, Pittsburgh); the average and standard deviation of each set of SVV estimations was calculated to assess accuracy and precision of intraindividual estimations, respectively.Ocular and cervical vestibular-evoked myogenic potentials (oVEMPs and c VEMPs) were registered using a commercial system with a 2-channel averaging capacity (ICS Chartr EP200, Otometrics, Taastrup, Denmark). Acoustic stimuli were 500 Hz tone bursts at 100 dBHL, delivered monaurally via headphones (TDH-49p, Telephonics, New York, NY, USA) at a rate of 5.1/s (2-0-2 ms rise/fall time; Blackman envelope, rarefaction polarity). To record ocular VEMPS (oVEMPs), participants were seated upright, and they were instructed to fixate on a target positioned 30° from their neutral gaze at 1.5 m; surface electrodes were positioned on the inferior oblique muscle of each eye, a reference electrode was positioned 2 cm below the orbital margin on the midline of the eye, and a ground electrode was positioned on the forehead. To record cervical VEMPs (cVEMPs), participants were seated upright, and they were instructed to maintain their heads turned contralaterally to the stimulated ear; surface electrodes were symmetrically positioned at the middle third of each sternocleidomastoid muscle, a reference electrode was positioned on the upper third of the sternum, and a ground electrode was positioned at the forehead. The impedance of the skin was maintained below 5 KOhms. For oVEMPs, the EMG signal was amplified (10 k gain) and bandpass filtered (0.1 Hz–1 kHz); the analysis window was 100 ms wide, and responses to 100 stimuli were averaged. For cVEMPs, the EMG signal was amplified (5 k gain) and bandpass filtered (1 Hz–1 kHz); the analysis window was 100 ms wide, and responses to 150 stimuli were averaged. For the purpose of this study, only presence/absence of a response was considered.

### 2.3. Statistical Analysis

Assessment of data distribution was performed using the Kolmogorov–Smirnov test. Accordingly, preliminary bivariate analysis was performed using either Mann–Whitney “U” test or “t” test (either for means or for proportions) with Holm–Bonferroni correction [41]. Simple correlations were assessed by the Spearman correlation coefficient, including the data of all the participants. According to the results of the bivariate analysis, multivariate analysis was performed using analysis of covariance (ANCoVA) on the sway recordings, the space anxiety scores and subscores, and the dizziness-related handicap scores and subscores, including the data of all the participants. All the tests were performed using Statistica (Statsoft, Tulsa, OK, USA) with a bilateral significance level of 0.05.

## 3. Results

### 3.1. Bivariate Analysis

Vestibular function test. Responses to sinusoidal rotation and the mean SVV deviation (accuracy) were similar in the two groups, and without asymmetries; while the intraindividual standard deviation of estimations (precision) of the SVV, (both static and during on-axis yaw rotation) was larger in patients with PPPD than in participants with no vestibular disease (Table 2).

Additionally, five patients with PPPD and a history of BPPV had no oVEMPS with no evidence of semicircular canal dysfunction on the sinusoidal rotation tests, two of them had a history of systemic high blood pressure and one of high cholesterol; also, two patients with PPPD and no history of specific vestibular diagnosis showed no oVEMPS and no cVEMPS, with decreased response to sinusoidal rotation in the dark just at 0.16 Hz rotation but effective response to rotation in the light at 0.16 Hz and in the dark at 1.28 Hz; one of them had a history of systemic high blood pressure. This small group of seven patients showed high variability on the evaluation questionnaires, within the range of the other patients participating in the study.

Posturography. Increased area and length of sway were evident for all conditions, which were the less for the eyes closed condition while standing on hard surface (Table 3); while the VFY showed a trend to decreased displacement in patients with PPPD (Table 3).

Questionnaires. The scores on the questionnaires are described on Table 4. As expected, significant differences between the two groups were evident on all the questionnaire scores and subscores. The symptoms of unsteadiness reported by at least 50% of the patients with PPPD, (apart from dizziness/vertigo) are shown in Figure 1; of note, the responses showed that instability was related to deficient sensory information and to movement of any of both the self or the environment. Consistently, the ten most frequent symptoms of depersonalization/derealization reported by the patients with PPPD (apart from dizziness) (Figure 2) included those related to perceptual deficits, such as “feel as if walking on shifting ground”, “difficulty understanding what others say to you”, and “vision is dulled”; however, almost all the patients reported difficulty concentrating and focusing attention (Figure 2).

Simple Correlation. Less consistent simple relationships were observed for the standard deviation of the static SVV than for the standard deviation of the SVV while on-axis yaw rotation SVV (Table 5). The standard deviation of the static SVV showed low correlation to the scores on unsteadiness, depersonalization/derealization symptoms, navigation subscore of spatial anxiety, and the total score and subscores on dizziness-related handicap (coefficients from 0.20 to 0.24, *p* ≤ 0.03). The standard deviation of the on-axis yaw rotation SVV showed low to moderate correlation to the scores on unsteadiness, anxiety subscore and total score of the HADS, the total score and subscores of spatial anxiety, and the total score and subscores on dizziness-related handicap (coefficients from 0.21 to 0.39, *p* ≤ 0.01). Inconsistent and low correlations were also observed between the standard deviation of the SVV, either static or while on-axis rotation, and the area, the length, and the VFY of the sway recordings.

### 3.2. Covariance Analysis

Posturography. The covariance analysis on the SVV standard deviation while on-axis yaw rotation to the posturography measurements was significant just for the length of sway and the VFY. Considering age and sex, contribution to the variance of the VFY was from 6% to 24% for all conditions (Table 6); while contribution to the variance of the length of sway was from 5% to 12% mainly in conditions on the hard surface (Table 6).

Space Anxiety inventory. The covariance analysis showed contributions to the variance of the spatial anxiety total score and all subscores from HADS total score ≥11 (inverse relationship), the standard deviation of the SVV while on-axis yaw rotation, and from sex (except from the Imaginery subscale), explaining 24% to 41% of the variance, regardless of the age and the score on perceived stress (Table 7).

Dizziness handicap inventory. The covariance analysis on the dizziness handicap inventory score and subscores showed contribution to the variance from the sway length while standing on the hard surface with the eyes open, either without/with neck extension, the navigation spatial anxiety subscore, the depersonalization/derealization score, and the absolute HADS subscore on anxiety, regardless of the age and sex (Table 8).

## 4. Discussion

In this study, patients with PPPD had increased variability (decreased precision) on SVV estimations, despite an average estimation (accuracy) similar to those of adults with no vestibular disease. Independent of the age of the participants, particularly when both otoliths and semicircular canals were stimulated, the precision decrease in estimating the force of gravity was related to sway recordings during a variety of conditions in upright stance; while patients with PPPD showed increased length of sway and decreased high-velocity compensating movements (VFY) mainly when vision was available. The results also showed that, including participants with/without PPPD, the precision of SVV estimation can be related to common mental symptoms, including derealization, and to spatial anxiety (especially on the navigation subdomain), emphasizing central processing.

Even in the absence of active behavior on Earth, the experience of inertial and gravitational reaction forces is a nexus among the sensory systems, allowing incorporation of one’s body into a conception of one-self in the environment [42]. In contrast, uncertain gravity perception may underlie inadequate postural control and emotional reactions. Previous reports have documented that, while standing upright, patients with PPPD may have difficulties with postural control across multiple sensory challenges [43]; they may be more dependent on visual input and less dependent on somatosensory input than healthy subjects [44]. This study supports that patients with PPPD can have uncertainty on gravity perception affecting their control of upright posture, with increased anxiety and unreality perceptions, which in turn contribute to dizziness-related handicap.

The results are coherent with the relevance of otolith information for SVV estimation in static conditions, and for navigation by path integration requirements for separation of the effects of tilt from those of translation by central processing [45]. Multiple cortical areas respond to vestibular stimuli, typically together with other sensory and motor signals (for review [46]) contributing to an internal representation of gravity effects based on prior experience [47]. Nevertheless, the thalamus may be fundamentally involved in the neural representation of a second graviceptive system for sensorimotor integration and to support adjustments to sudden changes in the environment [48,49].

Among the 53 patients participating in this study, seven (13%, 95% confidence interval 4 to 22%) showed absent oVEMPs which were mainly related to a history of BPPV. This finding aligns with recent reports of otolith dysfunction in patients with PPPD [19,20]. However, the reduced number of patients showing these findings and their variable results in the evaluations performed preclude any conclusion on the subject from this study.

Of note, the instruments administered in this study provided insight into the clinical manifestations of PPPD. The standardized questionnaire of symptoms of unsteadiness allowed easy reporting of the main symptoms depicting the disease (with no diagnostic purpose). The Depersonalization/Derealization Inventory provided the opportunity to assess distorted perceptions of the self and the environment, including sensory misperceptions and attention/concentrations deficits that were strongly related to the dizziness-related handicap, which is consistent with a previous report on derealization symptoms related to poor or incomplete recovery from peripheral vestibular disease [50]. The HADS total score and its anxiety subscore consistently showed the known relationship between persistent dizziness and mental morbidity [12]. The Spatial Anxiety Inventory showed that navigation spatial anxiety may have influence on the dizziness-related handicap, which usually prevails in women [38,51]. The impact of navigation spatial anxiety on the dizziness-related handicap could be expected since navigation ability is essential in daily life, with an impact on autonomy [52]. Although the Perceived Stress Scale score was different between the two study groups, it showed no correlation with the spatial anxiety score/subscores reported by the patients with PPPD, in contrast to the relationship observed in patients with active peripheral vestibular disorders [29], which agrees with dissimilar processes giving rise to spatial anxiety in these two groups of patients.

Consistent with the strong contribution of anxiety to the variance on dizziness-related handicap (Table 8), imaging studies support that patients with PPPD and patients with anxiety disorders may share anxiety neuronal networks, with decreased adaptation to emotional stimuli [53], while space–motion discomfort has been associated with increased reliance on visual cues for postural control in patients with anxiety disorders [54]. On the other hand, symptoms of anxiety, but not of depression, were inversely related to spatial anxiety (Table 7), a finding that is similar to the observations on patients with a variety of vestibular disorders [29], and that is consistent with evidence showing that predisposition for emotional reactions could be helpful to overcome the apprehension provoked by a spatial task [55] but requires further revision and intended studies.

The main limitation of this study is its cross-sectional design that prevents discussion on any causal relationship. A second limitation is the stringent selection criteria for participation, which was required to evaluate a limited number of factors; then, the results may not accurately represent the broader PPPD population; however, this selective approach allowed for a focused analysis. The enrolment was performed in a specialized clinic to include patients with a first-time diagnosis. The results could be different in other clinical settings, and intended studies should be performed to distinguish responses between patients with/without a history of vestibular disease; in addition, according to the purpose of the study, participants without PPPD were selected to have no sensory dysfunction (other than refractive errors), future studies including participants with other sensory deficits would be valuable. Since the questionnaires were administered for self-report, inaccurate information might have been included in the analysis. The main strengths of this study were that it was performed by trained health professionals, who collected the data prospectively in a standardized manner, while assessments included a variety of cofactors, and the results were consistent within the same study and with previous reports. Another limitation is that this study did not allow assessment of a potential influence on the results from decision making under uncertainty. This potential influence was not anticipated in the study design, since no previous report on SVV precision in patients with PPPD or data suggesting its influence on postural control (without visuo-vestibular conflict) were identified in the literature. Replication of the results in a variety of experimental conditions will be relevant to ascertain the findings.

## 5. Conclusions

The results suggest that uncertain gravity perception could be a contributing factor to postural instability, as well as to misperceptions of reality and spatial anxiety, which in turn contribute to the perceived dizziness-related handicap reported by patients with PPPD.

## Figures and Tables

**Figure 1 jcm-13-06665-f001:**
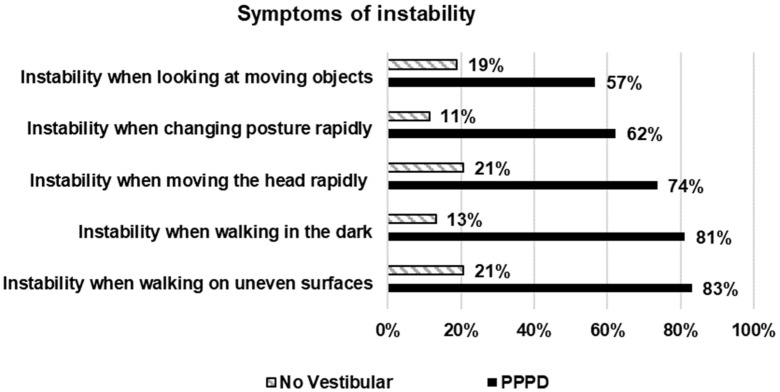
Frequency of symptoms of unsteadiness apart from dizziness/vertigo that were reported by at least 50% of the 53 patients with Persistent Postural Perceptual Dizziness, and the frequency on 53 participants with no vestibular disease. Statistical difference between the two groups was significant for all the comparisons (*t* test for proportions, *p* < 0.05).

**Figure 2 jcm-13-06665-f002:**
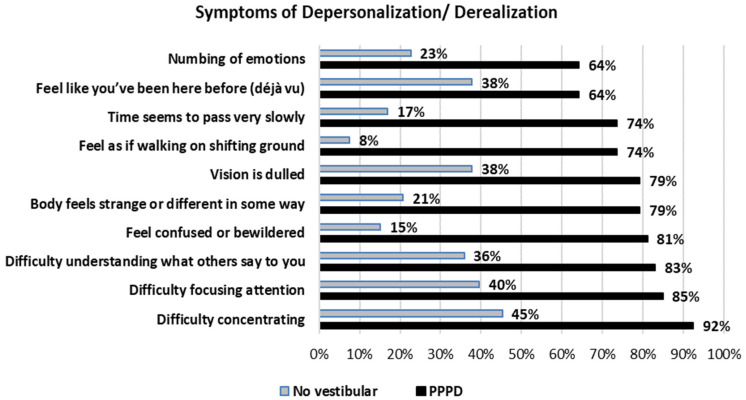
The 10 most frequent symptoms of depersonalization/derealization reported by 53 patients with Persistent Postural Perceptual Dizziness and 53 participants with no vestibular disease. Statistical difference between the two groups was significant for all the comparisons (*t* test for proportions, *p* < 0.05).

**Table 1 jcm-13-06665-t001:** General characteristics of the participants.

Variable	No Vestibular Disease	PPPD
Number of participants	53	53
Men/Women ratio	13/40	13/40
Years of age (mean ± S.D.)	53.1 ± 11.0	52.5 ± 11.7
Body mass index (mean ± S.D.)	28.0 ± 3.8	29.4 ± 4.4
Years at school (mean ± S.D.)	12.4 ± 3.6	11.9 ± 3.1
Spectacles due to refraction errors (N,%)	44 (83%)	46 (86%)
Tobacco smokers (N, %)	5 (9%)	6 (11%)
Alcohol use (N, %)	14 (26%)	8 (15%)
Comorbidities		
Diabetes (N, %)	2 (3%)	0
Systemic high blood pressure (N, %)	10 (18%)	12 (22%)
Diabetes and high blood pressure (N, %)	0	5 (9%)
Other (N, %)	3 (5%)	7 (13%)
Months since last spinning vertigo (median, Q1–Q3)	-	4 (4–5)

**Table 2 jcm-13-06665-t002:** Mean and standard deviation of the mean of the vestibular tests on 53 participants with no vestibular disease and 53 patients with Persistent Postural Perceptual Dizziness. Comparisons were performed using “*t*” test (df = degrees of freedom) (* denotes significance after Holm–Bonferroni correction).

Test	No Vestibular Disease	PPPD	*p* (t Value)
	(n = 53)	(n = 53)	df 104
Gain to sinusoidal rotation (mean ± S.D.)			
Visual fixation at 0.16 Hz	0.07 ± 0.03	0.06 ± 0.03	0.32 (0.99)
In the light at 0.16 Hz	0.97 ± 0.08	0.97 ± 0.08	0.89 (0.13)
In the dark at 0.16 Hz	0.52 ± 0.11	0.52 ± 0.14	0.16 (0.87)
In the dark at 1.28 Hz	0.99 ± 0.07	0.97 ± 0.07	0.25 (1.14)
Subjective Visual Vertical (mean ± S.D)			
Static			
Average (accuracy)	0.13 ± 0.96	0.21 ± 0.84	0.64 (0.46)
Standard deviation (precision)	0.79 ± 0.46	1.08 ± 0.55	0.01 (2.86) *
On-axis yaw rotation			
Average (accuracy)	0.08 ± 0.86	0.28 ± 0.70	0.19 (1.31)
Standard deviation (precision)	0.76 ± 0.51	1.19 ± 0.74	0.0008 (3.45) *

**Table 3 jcm-13-06665-t003:** Median and quartiles 1 and 3 (Q1–Q3) of the sway recording of 53 participants with no vestibular disease and 53 patients with Persistent Postural Perceptual Dizziness (PPPD). Comparisons were performed using Mann–Whitney “U” test (df = degrees of freedom) (* denotes significance after Holm–Bonferroni correction).

Condition	No Vestibular Disease	PPPD	*p* (Z Value)
	(n = 53)	(n = 53)	df 104
Area of sway (mm^2^)			
Hard surface			
Open eyes	81 (55–120)	109 (81–187)	0.003 (2.9) *
Closed eyes	111 (60–193)	145 (99–326)	0.04 (2.0)
No neck extension/open eyes	71 (37–101)	91 (55–225)	0.003 (2.9) *
Neck extension/open eyes	62 (45–102)	94 (64–227)	0.004 (2.8) *
Soft surface			
Open eyes	126 (84–179)	153 (106–325)	0.01 (2.4) *
Closed eyes	268 (176–389)	372 (211–564)	0.04 (1.9)
No neck extension/open eyes	116 (72–187)	194 (105–305)	0.002 (3.0) *
Neck extension/open eyes	125 (92–177)	181 (103–387)	0.007 (2.6) *
Length of sway (mm)			
Hard surface			
Open eyes	319 (263–364)	387 (309–436)	0.003 (2.8) *
Closed eyes	426 (349–543)	535 (389–667)	0.03 (2.1)
No neck extension/open eyes	283 (237–335)	352 (271–429)	0.002 (3.0) *
Neck extension/open eyes	288 (255–379)	387 (302–502)	0.001 (3.1) *
Soft surface			
Open eyes	367 (299–467)	431 (359–480)	0.02 (2.17)
Closed eyes	623 (484–721)	699 (523–851)	0.03 (2.0)
No neck extension/open eyes	355 (268–442)	415 (335–544)	0.004 (2.8) *
Neck extension/open eyes	374 (319–443)	439 (346–564)	0.005 (2.7) *
Velocity displacement in the sagittal plane			
Hard surface			
Open eyes	−8.6 (−10–−6.9)	−7.5 (−10.4–−5.1)	0.07 (1.7)
Closed eyes	−4.5 (−9–−1.9)	−3.6 (−6.5–1.3)	0.09 (1.6)
No neck extension/open eyes	−8.9 (−11–−7.2)	−8.0 (−9.5–−5.2)	0.03 (2.1)
Neck extension/open eyes	−8.6 (−11–−6.1)	−7.1 (−9.8–−1.8)	0.04 (2.0)
Soft surface			
Open eyes	−9.5 (−10–−8.1)	−8.2 (−10.6–−6.9)	0.03 (2.0)
Closed eyes	−1.6 (−6–1.9)	1.0 (−4.3–5.1)	0.04 (2.0)
No neck extension/open eyes	−10.2 (−11–−8.8)	−8.6(−10.8–−6.4)	0.01 (2.4) *
Neck extension/open eyes	−7.4 (−12–−3.8)	−6.8 (−10.5–−0.3)	0.41 (1.4)

**Table 4 jcm-13-06665-t004:** Median and quartiles 1 and 3 (Q1–Q3) of the scores on the questionnaires administered to 53 participants with no vestibular disease and 53 patients with Persistent Postural Perceptual Dizziness (PPPD). Comparisons were performed using Mann–Whitney “U” test (df = degrees of freedom) (* denotes significance after Holm–Bonferroni correction).

Instrument	No Vestibular Disease	PPPD	*p* (Z Value)
	(n = 53)	(n = 53)	(df 104)
Symptoms related to unsteadiness	1 (0–2)	6 (5–7)	<0.00001 (8.9) *
Hospital Anxiety and Depression Scale			
Anxiety subscore	3 (2–7)	9 (7–14)	<0.00001 (6.1) *
Depression subscore	2 (0–4)	6 (4–9)	<0.00001 (4.9) *
Total score	5 (3–10)	16 (11–22)	<0.00001 (6.2) *
Perceived Stress Scale-10	12 (8–16)	20 (18–23)	<0.00001 (5.4) *
Depersonalization/Derealization Inventory	3 (1–8)	28 (15–23)	<0.00001 (8.0) *
Spatial Anxiety Scale			
Spatial imagery subscore	5 (2–11)	14 (9–17)	<0.00001 (5.2) *
Spatial navigation subscore	6 (3–12)	18 (13–21)	<0.00001 (6.2) *
Spatial mental manipulation subscore	5 (2–15)	15 (9–19)	<0.00001 (4.3) *
Total score	16 (8–42)	45 (36–55)	<0.00001 (5.5) *
Dizziness Handicap Inventory			
Functional subscore	0 (0–0)	16 (10–20)	<0.00001 (8.5) *
Physical subscore	0 (0–0)	14 (10–18)	<0.00001 (8.9) *
Emotional subscore	0 (0–0)	12 (6–18)	<0.00001 (8.3) *
Total score	0 (0–0)	44 (26–58)	<0.00001 (8.7) *

**Table 5 jcm-13-06665-t005:** Spearman rank correlation between the standard deviation of the Subjective Visual Vertical (either static and during on-axis rotation) and the evaluation questionnaires of 106 participants, 53 with and 53 without Persistent Postural Perceptual Dizziness (df = degrees of freedom) (significant coefficients are highlighted using * *p* ≤ 0.05 and ** *p* ≤ 0.005).

	Static SVV		On-Axis SVV	
Instrument	Coefficient	*p* (t Value, df 104)	Coefficient	*p* (t Value, df 104)
Symptoms related to unsteadiness	0.21 *	0.02 (2.26)	0.39 **	0.005 (4.35)
Hospital Anxiety and Depression Scale				
Anxiety subscore	0.09	0.31 (1.01)	0.20	0.03 (2.1)
Depression subscore	0.04	0.66 (0.43)	0.16	0.09 (1.7)
Total score	0.07	0.46 (0.73)	0.21	0.03 (2.1)
Perceived Stress Scale-10	0.15	0.11 (1.56)	0.15	0.11 (1.5)
Depersonalization/Derealization Inventory	0.20 *	0.03 (2.15)	0.37 **	<0.00001 (4.0)
Spatial Anxiety Scale				
Spatial imagery subscore	0.18	0.057 (1.91)	0.24 *	0.01 (2.6)
Spatial navigation subscore	0.22 *	0.02 (2.35)	0.27 **	0.004 (2.9)
Spatial mental manipulation subscore	0.14	0.14 (1.47)	0.27 **	0.004 (2.9)
Total score	0.18	0.052 (1.96)	0.27 **	0.004 (2.9)
Dizziness Handicap Inventory				
Functional subscore	0.22 *	0.01 (2.38)	0.32 **	0.0008 (3.4)
Physical subscore	0.20 *	0.03 (2.17)	0.34 **	0.0003 (3.7)
Emotional subscore	0.24 *	0.01 (2.62)	0.34 **	0.0002 (3.8)
Total score	0.21 *	0.02 (2.29)	0.34 **	0.0004 (3.6)

**Table 6 jcm-13-06665-t006:** Adjusted R^2^, beta values with 95% confidence interval of the age, the subjective visual vertical (SVV) precision during on-axis rotation, and sex included in the general linear model on the length of sway and the velocity displacement of the center of pressure in the sagittal plane (VFY) during the posturography conditions (df = degrees of freedom).

	LENGTH OF SWAY			VFY		
CONDITION Factors	Adjusted R^2^ *p*, F (df)	Beta	95% C.I.	Adjusted R^2^ *p*, F (df)	Beta	95% C.I.
HARD SURFACE						
Open eyes	0.05 0.03, 2.9 (8, 95)			0.07 0.009, 4.0 (8, 95)		
Age		0.23	0.03–0.42		0.11	−0.07–0.30
SVV standard deviation		0.17	−0.01–0.35		0.16	−0.02–0.35
Sex		−0.06	−0.26–0.12		0.24	0.05–0.43
Closed eyes	0.07 0.009, 4.0 (8, 95)			0.06 0.01, 3.5 (8, 95)		
Age		0.05	−0.13–0.24		0.03	−0.15–0.22
SVV standard deviation		0.26	0.07–0.44		0.30	0.11–0.49
Sex		−0.18	−0.37–0.008		−0.02	−0.22–0.16
No neck extension/open eyes	0.07 0.009, 4.02 (8, 95)			0.14 0.0002, 6.8 (8, 95)		
Age		0.16	−0.02–0.36		0.09	−0.09–0.27
SVV standard deviation		0.26	0.07–0.45		0.31	0.13–0.49
Sex		−0.11	−0.30–0.07		0.24	0.06–0.43
Neck extension/open eyes	0.12 0.0009, 5.8 (8, 95)			0.10 0.002, 5.1 (8, 95)		
Age		0.14	−0.04–0.32		0.10	−0.08–0.28
SVV standard deviation		0.36	0.17–0.54		0.32	0.14–0.50
Sex		−0.01	−0.20–0.17		0.13	−0.05–0.32
SOFT SURFACE						
Open eyes	0.12 0.0007, 6.0 (8, 95)			0.24 <0.00001, 9.9 (8, 95)		
Age		0.30	0.12–0.49		0.12	−0.04–0.30
SVV standard deviation		0.23	0.05–0.41		0.21	0.04–0.38
Sex		−0.15	−0.33–0.032		0.39	0.21–0.56
Closed eyes	0.08 0.007, 4.2 (8, 95)			0.06 0.02, 3.2 (8, 95)		
Age		−0.01	−0.20–0.17		−0.12	−0.32–0.06
SVV standard deviation		0.23	0.04–0.41		0.25	0.06–0.44
Sex		−0.22	−0.41–−0.03		0.13	−0.06–0.32
No neck extension/open eyes	0.04 0.054, 2.6 (8, 95)			0.18 0.00002, 8.8 (8, 95)		
Age		0.15	−0.03–0.34		0.10	−0.07–0.28
SVV standard deviation		0.20	0.01–0.39		0.16	−0.006–0.34
Sex		−0.10	−0.30–0.08		0.39	0.21–0.57
Neck Extension/open eyes	0.05 0.03, 3.0 (8, 95)			0.06 0.02, 3.2 (8, 95)		
Age		0.13	−0.05–0.33		0.05	−0.13–0.25
SVV standard deviation		0.25	0.06–0.44		0.22	0.03–0.41
Sex		−0.04	−0.23–0.15		0.18	−0.005–0.37

**Table 7 jcm-13-06665-t007:** Adjusted R^2^, beta values and 95% confidence interval (C.I.) of the beta values of the variables included in the general linear model on the Space Anxiety Score and subscores (SVV = Subjective Visual Vertical; HADS = Hospital Anxiety and Depression Scale; df = degrees of freedom).

Subscore	Imaginery		Navigation		Mental Manipulation		Total	
Adjusted R^2^ *p*, F(df)	0.24 <0.00001, 6.7 (3, 97)		0.41 <0.00001, 13.2 (3, 97)		0.30 <0.00001, 8.5 (3, 97)		0.38 <0.00001, 11.8 (3, 97)	
Factors	Beta	95% C.I.	Beta	95% C.I.	Beta	95% C.I.	Beta	95% C.I.
Age	0.12	−0.04–0.30	−0.01	−0.16–0.14	0.02	−0.14–0.19	0.04	−0.11–0.20
On-axis SVV Standard deviation	0.17	0.003–0.35	0.21	0.06–0.37	0.19	0.03–0.36	0.21	0.05–0.37
Perceived stress	0.08	−0.12–0.29	0.13	−0.05–0.31	−0.007	−0.20–0.19	0.07	−0.11–0.26
Sex	0.13	−0.03–0.30	0.19	0.03–0.34	0.23	0.06–0.40	0.20	0.04–0.36
Total HADS ≥ 11	−0.41	−0.64–−0.18	−0.52	−0.72–−031	−0.50	−0.72–−0.28	−0.52	−0.73–−0.31
Sex × total HADS ≥ 11	0.08	−0.10–0.28	0.13	−0.03–0.30	0.10	−0.08–0.29	0.11	−0.05–0.29

**Table 8 jcm-13-06665-t008:** Adjusted R^2^, beta values and 95% confidence interval (C.I.) of the beta values of the variables included in the general linear model on the Dizziness Handicap Inventory Scores and subscores, including (A) the length of sway while standing on hard surface with the eyes open, or (B) with 30° neck extension while standing on the hard surface with the eyes open (HADS = Hospital Anxiety and Depression Scale) (df = degrees of freedom).

A	Physical		Functional		Emotional		Total	
Adjusted R^2^ *p*, F (df)	0.59 <0.00001, 26.5 (3, 97)		0.63 <0.00001, 31.6 (3, 97)		0.59 <0.00001, 26.3 (3, 97)		0.66 <0.00001, 35.5 (3, 97)	
Factors	Beta	95% C.I.	Beta	95% C.I.	Beta	95% C.I.	Beta	95% C.I.
Age	0.009	−0.12–0.14	0.07	−0.04–0.20	0.01	−0.12–0.14	0.03	−0.08–0.15
Sway length (hard surface/open eyes)	0.20	0.06–0.33	0.17	0.05–0.30	0.17	0.04–0.30	0.19	0.07–0.31
Navigation spatial anxiety	0.11	−0.03–0.27	0.19	0.05–0.34	−0.05	−0.20–0.10	0.09	−0.04–0.23
Depersonalization/Derealization	0.51	0.32–0.71	0.48	0.30–0.66	0.51	0.32–0.70	0.53	0.35–0.70
Anxiety HADS subscore	0.15	−0.01–0.33	0.18	0.01–0.34	0.29	0.11–0.47	0.21	0.05–0.37
Sex	0.01	−0.11–0.15	0.007	−0.11–0.13	0.06	−0.07–0.19	0.03	−0.09–0.15
B								
Adjusted R^2^ *p*, F (df)	0.57 <0.00001, 24.8 (3, 97)		0.66 <0.00001, 32.4 (3, 97)		0.57 <0.00001, 24.2 (3, 97)		0.64 <0.00001, 33.3 (3, 97)	
Factors	Beta	95% C.I.	Beta	95% C.I.	Beta	95% C.I.	Beta	95% C.I.
Age	0.03	−0.09–0.17	0.09	−0.03–0.21	0.04	−0.08–0.18	0.06	−0.06–0.18
Sway length (hard surface/neck extension)	0.14	0.01–0.28	0.19	0.07–0.32	0.08	−0.05–0.22	0.15	0.02–0.27
Navigation spatial anxiety	0.09	−0.06–0.25	0.16	0.02–0.31	−0.06	−0.22–0.09	0.06	−0.07–0.21
Depersonalization/Derealization	0.53	0.34–0.73	0.47	0.29–0.65	0.54	0.35–0.74	0.54	0.37–0.72
Anxiety HADS subscore	0.16	−0.01–0.34	0.19	0.02–0.35	0.30	0.11–0.48	0.22	0.06–0.39
Sex	0.01	−0.12–0.14	0.006	−0.11–0.13	0.05	−0.07–0.19	0.02	−0.09–0.15

## Data Availability

The raw data supporting the conclusions of this article will be made available by the authors on request.

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
