# Peer review of "Graviception Uncertainty, Spatial Anxiety, and Derealization in Patients with Persistent Postural-Perceptual Dizziness"

_jcm, 2024, doi:10.3390/jcm13226665_

Round 1
Reviewer 1 Report
Comments and Suggestions for Authors
GENERAL: This is a very detailed, well referenced and well discussed study. I am not sure that this much detail is necessary, but the authors’ covering of what is understood in the literature is carried out well.
Authors have carried out a number of detailed vestibular assessments, which adds to the strength of their findings. They have illustrated a good knowledge of how to carry out these assessments and what the results mean.
The figures they have included are helpful and illustrate their findings well.
The discussion is also very detailed, but is well researched and well written. Again I will defer to the editors with respect to whether this much detail is appropriate.
The conclusion is succinct and is supported by their experimental finings
There are a few English corrections that should be made.
Line 13 – should read “vestibular patients agreed to participate”/
Line 45 - should be changed to use visual dependency to maintain balance
Line 103-104 - should be changed to there were 53 patients diagnosed with PPPD (25 to 84 years old; 40 women and 13 men),
Lines 169-187 are excessively detailed and I think this detail about methodology should be tightened up. A suggestion for this would be:
Static posturography was carried out using 8 different sensory conditions” (and then listing the 8 conditions)
Line 215 should read “for the purpose of this study”
Line 329 should say SVV estimations, despite an average estimation (accuracy) similar to those of adults with no vestibular disease. Independent of the age of the participants, particularly when both
Line 348 should say The results are consistent with and supportive of the use of otolith information for SVV estimation in static conditions, and for navigation by path integration requirements for separation of
The bibliography should end on line 567, (i.e. deleting all of the examples that are in the template)
Author Response
GENERAL: This is a very detailed, well referenced and well discussed study. I am not sure that this much detail is necessary, but the authors’ covering of what is understood in the literature is carried out well. Authors have carried out a number of detailed vestibular assessments, which adds to the strength of their findings. They have illustrated a good knowledge of how to carry out these assessments and what the results mean. The figures they have included are helpful and illustrate their findings well. The discussion is also very detailed but is well researched and well written. Again, I will defer to the editors with respect to whether this much detail is appropriate. The conclusion is succinct and is supported by their experimental finings
- We thank the reviewer for all these positive comments and edited the Introduction and Discussion to shorten them as recommended.
There are a few English corrections that should be made.
-We thank the reviewer for the corrections. All have been performed accordingly. They are highlighted in black lettering.
Line 13 – should read “vestibular patients agreed to participate”/
Line 45 - should be changed to use visual dependency to maintain balance
Line 103-104 - should be changed to there were 53 patients diagnosed with PPPD (25 to 84 years old; 40 women and 13 men),
Lines 169-187 are excessively detailed and I think this detail about methodology should be tightened up. A suggestion for this would be: Static posturography was carried out using 8 different sensory conditions” (and then listing the 8 conditions)
Line 215 should read “for the purpose of this study”
Line 329 should say SVV estimations, despite an average estimation (accuracy) similar to those of adults with no vestibular disease. Independent of the age of the participants, particularly when both
Line 348 should say The results are consistent with and supportive of the use of otolith information for SVV estimation in static conditions, and for navigation by path integration requirements for separation of
The bibliography should end on line 567, (i.e. deleting all of the examples that are in the template).
- We thank the reviewer for the observation. The lines have been deleted.
Reviewer 2 Report
Comments and Suggestions for Authors
◎Introduction
●The introduction is redundant. It is necessary to briefly state why the research topic was chosen and why the method/approach was adopted. It is necessary to explain the research questions that have been considered and the research objectives that have been set, and to briefly state the solutions and approaches that have not been presented in previous studies. If necessary, additional explanations should be provided in the Methods section and in the Discussion section.
◎Materials and Methods
●There is a lack of information on the assessment of vestibular function in the PPPD population. Posutrographic findings may be influenced by vestibular dysfunction or vestibular disease preceding PPPD. In addition to cVEMP and oVEMP findings (normal, unilateral dysfunction, bilateral dysfunction), information on vHIT (or caloric test) findings (normal, unilateral dysfunction, bilateral dysfunction) is needed.
◎Results
●Table 6
Table 6 is very difficult to read. In particular, the relationship between the headings Hard surface and Soft surface and the condition factor is difficult to read.
●Lines 298-300
The covariance analysis on the SVV standard deviation while on-axis yaw rotation to the posturography measurements was significant just for the length of sway and the VFY.
Where are the results of this analysis described? It is not clear whether these are the results of healthy subjects or PPPD patients.
●Lines 300-302
Considering age and sex, contribution to the variance of the VFY was 300 from 6 % to 24% for all conditions (Table 6); while contribution to the variance of the length 301 of sway was from 5% to 12% mainly in conditions on hard surface (Table 6).
It is not described whether the results in Table 6 are for PPPD patients. It is not clear whether this result is not observed in controls but only in the PPPD patient group. If the finding is common to controls and PPPD patients, it cannot be said to be characteristic of PPPD patients. It must be proven that this result is only observed in PPPD patients.
●Lines 310-314
The covariance analysis showed contributions to the vari-310 ance of the spatial anxiety total score and all sub-scores from HADS total score ≥11 (in-311 verse relationship), the standard deviation of the SVV while on-axis yaw rotation, and 312 from sex (except from the Imaginery subscale), explaining 24% to 41% of the variance, 313 regardless of the age and the score on perceived stress (Table 7).
It is not described whether the results in Table 7 are for PPPD patients. It is not clear whether this result is not observed in controls but only in the PPPD patient group. If the finding is common to controls and PPPD patients, it cannot be said to be characteristic of PPPD patients. It must be proven that this result is only observed in PPPD patients.
●Lines 315-319:
The covariance analysis on the dizziness handicap in-315 ventory score and sub-scores showed contribution to the variance from the sway length 316 while standing on hard surface with the eyes open, either without/with neck extension, 317 the navigation spatial anxiety subscore, the depersonalization/ derealization score and the 318 absolute HADS subscore on anxiety, regardless of the age and sex (Table 8).
It is not described whether the results in Table 8 are for PPPD patients. It is not clear whether this result is not observed in controls but only in the PPPD patient group. If the finding is common to controls and PPPD patients, it cannot be said to be characteristic of PPPD patients. It must be proven that this result is only observed in PPPD patients.
●The diagnostic criteria require that patients with PPPD have a preceding dizziness-causing condition. Although the symptoms of patients with PPPD cannot be explained by preceding condition, examination findings may be influenced by the preceding condition. Swaying in upright posture and SVV finding may be affected if the preceding dizziness-causing condition is vestibular disease, for example. Is it not possible that the variability of SVV in PPPD could be looking at the influence of otolith organ function in the patient group, and that the association between SVV variability and the length of posturography postural sway could be an effect of the high prevalence of vestibular dysfunction in the PPPD patient group?
Author Response
Introduction
The introduction is redundant. It is necessary to briefly state why the research topic was chosen and why the method/approach was adopted. It is necessary to explain the research questions that have been considered and the research objectives that have been set, and to briefly state the solutions and approaches that have not been presented in previous studies. If necessary, additional explanations should be provided in the Methods section and in the Discussion section.
- We thank the reviewer for eth comment, the introduction has been simplified and the aim of the study is described in the final paragraph.
Materials and Methods
There is a lack of information on the assessment of vestibular function in the PPPD population. Posutrographic findings may be influenced by vestibular dysfunction or vestibular disease preceding PPPD. In addition to cVEMP and oVEMP findings (normal, unilateral dysfunction, bilateral dysfunction), information on vHIT (or caloric test) findings (normal, unilateral dysfunction, bilateral dysfunction) is needed.
- We agree with the reviewer that the preceding disease may have influenced the results. However, according to the Barany Society criteria, it was diagnosed months before participating in the study, as it is described in the Methods section (participants subsection 2.1): “ 10 patients had a previous diagnosis of unilateral vestibular dysfunction and 20 of Benign Paroxysmal Positional Vertigo (BPPV), while the remaining 23 had no specific vestibular diagnosis but reported a previous history of dizziness/vertigo”,
- The results of the vestibular tests performed in the study are described in the Results section (subsection 3.1) and in Table 2; of note, the description has been edited to clarify that the responses to rotation were symmetrical. Apart from the patient with decreased symmetrical response to rotation at 0.16 Hz, but normal response at 1.28 Hz, no other abnormalities were observed.
- In the two groups Sinusoidal rotation at 0.16 Hz and at 1.28 Hz (60°/s peak velocity) was performed to assess the VOR gain, no caloric tests were performed. The following results for each group are described in Table 2:
Test |
No vestibular disease |
PPPD |
p (t value) |
|
(n=53) |
(n=53) |
df 104 |
Gain to sinusoidal rotation (mean ± S.D.) |
|
|
|
Visual Fixation at 0.16 Hz |
0.07 ± 0.03 |
0.06 ± 0.03 |
0.32 (0.99) |
In the light at 0.16 Hz |
0.97 ± 0.08 |
0.97 ± 0.08 |
0.89 (0.13) |
In the dark at 0.16 Hz |
0.52 ± 0.11 |
0.52 ± 0.14 |
0.16 (0.87) |
In the dark at 1.28 Hz |
0.99 ± 0.07 |
0.97 ± 0.07 |
0.25 (1.14) |
Results
Table 6. Table 6 is very difficult to read. In particular, the relationship between the headings Hard surface and Soft surface and the condition factor is difficult to read.
- We thank the reviewer for the observation. The Table was revised and the format was updated to facilitate reading.
Lines 298-300: “The covariance analysis on the SVV standard deviation while on-axis yaw rotation to the posturography measurements was significant just for the length of sway and the VFY.” Where are the results of this analysis described? It is not clear whether these are the results of healthy subjects or PPPD patients.
- We thank the reviewer for the comment. It has been clarified that the analysis included the full range of data (subsection 2.3), to prevent inflating the correlations.
- To assess correlation, it is advisable to include the full range of data. An important pitfall of the correlation coefficient is that it is influenced by the range of observations; including mainly high or low values could inflate the correlation coefficients.
- The results are described in section 3.2, all details are described in Table 6, including the adjusted multiple R2 (with p value, F value and degrees of freedom), as well as the beta values ( with the 95% Confidence Intervals).
Comments on the results described in Tables 6, 7 and 8:
Lines 300-302: “Considering age and sex, contribution to the variance of the VFY was from 6 % to 24% for all conditions (Table 6); while contribution to the variance of the length of sway was from 5% to 12% mainly in conditions on hard surface (Table 6). “ It is not described whether the results in Table 6 are for PPPD patients. It is not clear whether this result is not observed in controls but only in the PPPD patient group. If the finding is common to controls and PPPD patients, it cannot be said to be characteristic of PPPD patients. It must be proven that this result is only observed in PPPD patients.
Lines 315-319: “The covariance analysis on the dizziness handicap in ventory score and sub-scores showed contribution to the variance from the sway length while standing on hard surface with the eyes open, either without/with neck extension, the navigation spatial anxiety subscore, the depersonalization/ derealization score and the absolute HADS subscore on anxiety, regardless of the age and sex (Table 8).” It is not described whether the results in Table 8 are for PPPD patients. It is not clear whether this result is not observed in controls but only in the PPPD patient group. If the finding is common to controls and PPPD patients, it cannot be said to be characteristic of PPPD patients. It must be proven that this result is only observed in PPPD patients.
Lines 310-314: “The covariance analysis showed contributions to the variance of the spatial anxiety total score and all sub-scores from HADS total score ≥11 (inverse relationship), the standard deviation of the SVV while on-axis yaw rotation, and from sex (except from the Imaginery subscale), explaining 24% to 41% of the variance, regardless of the age and the score on perceived stress (Table 7).“ It is not described whether the results in Table 7 are for PPPD patients. It is not clear whether this result is not observed in controls but only in the PPPD patient group. If the finding is common to controls and PPPD patients, it cannot be said to be characteristic of PPPD patients. It must be proven that this result is only observed in PPPD patients.
- The study aim was “to assess the correlation between the accuracy and precision of SVV with (1) the postural sway in static upright-position in varied conditions, and with (2) spatial anxiety (three domains), and (3) their impact on the dizziness-related handicap reported by adults with PPPD …”. To prevent bias on correlation estimation,, it is advisable to include the full range of data. An important pitfall of the correlation coefficient is that it is influenced by the range of observations; including mainly high or low values could inflate the correlation coefficients.
- Since we advocate for the dysfunction of physiological processes, we expected the correlation to be evident in the full range of data.
The diagnostic criteria require that patients with PPPD have a preceding dizziness-causing condition. Although the symptoms of patients with PPPD cannot be explained by preceding condition, examination findings may be influenced by the preceding condition. Swaying in upright posture and SVV finding may be affected if the preceding dizziness-causing condition is vestibular disease, for example. Is it not possible that the variability of SVV in PPPD could be looking at the influence of otolith organ function in the patient group, and that the association between SVV variability and the length of posturography postural sway could be an effect of the high prevalence of vestibular dysfunction in the PPPD patient group?
-Participants with PPPD fulfilled the Barany Society criteria and were evaluated in a specialized clinic before participating in the study; later, during participation, the study assessments showed no evidence of acute disease.
- The study design was cross-sectional; it cannot ascertain any causal relationship but support future studies.
Reviewer 3 Report
Comments and Suggestions for Authors
This study evaluated the influence of graviception, anxiety and several scores on persistent postural perceptual dizziness (PPPD) and found that unprecise graviception may be a contributing factor. As is often the case with papers on this topic, there is a problem in that it is unclear whether the control group can be accurately called a control, and it should be described in detail as a limitation.
There are several grammatical errors, so all texts should be checked again thoroughly.
L112 Society.12 (12?)
L116 dizziness/ vertigo (spacing error)
L117 ear/ neurological/ … (spacing error)
L118 dB nHL The hearing threshold of the pure tone audiometry is expressed in dBHL.
Table 1. Alcohol use Why PPPD patients drink less alcohol?
L170 16 b (bits?)
L197 100 dB HL The sound pressure of VEMP should be expressed in dB peSPL or dBpSPL.
L225 0.05.. (.?)
Table 6
Some lines are missing. There are some spacing errors (ex. 4.02(8,95) no space).
Table 7, 8
Some lines are missing.
L375 L377 / spacing errors
L437 explanation (?)
Author Response
This study evaluated the influence of graviception, anxiety and several scores on persistent postural perceptual dizziness (PPPD) and found that unprecise graviception may be a contributing factor. As is often the case with papers on this topic, there is a problem in that it is unclear whether the control group can be accurately called a control, and it should be described in detail as a limitation.
- We thank the reviewer for the comments. The section 2.1 of the Methods has been edited to state that : No evidence of vestibular dysfunction was observed in participants without PPPD on their clinical records and the vestibular evaluations performed to participate in the study.
- The limitations paragraph was also edited, to include that “… according to the purpose of the study, participants without PPPD were selected to have no sensory dysfunction (other than refractive errors), future studies including participants with other sensory deficits would be valuable.”
There are several grammatical errors, so all texts should be checked again thoroughly.
- We thank the reviewer for all the corrections, which have been performed accordingly.
L112 Society.12 (12?)
- We thank the reviewer for the observation. The reference is now properly written: [12]
L116 dizziness/ vertigo (spacing error)
- Thank you, the error has been corrected.
L117 ear/ neurological/ … (spacing error)
- Thank you, the error has been corrected.
L118 dB nHL The hearing threshold of the pure tone audiometry is expressed in dBHL.
- Thank you, the unit’s description has been corrected.
Table 1. Alcohol use Why PPPD patients drink less alcohol?
- We have not speculated on the finding. However, we may assume that they have gotten medical advice; while they are usually unwell and may not want to increase that felling.
L170 16 b (bits?)
- Yes, bits. However, the paragraph has been edited with deletion for simplicity, as requested by Reviewer 1.
L197 100 dB HL The sound pressure of VEMP should be expressed in dB peSPL or dBpSPL.
- Thank you. We used an equipment providing stimuli on dBHL (Otometrics ICS Chartr EP 200 Installation And Startup Manual, page 36).
L225 0.05.. (.?)
- Thank you, the double stop has been deleted.
Table 6
Some lines are missing. There are some spacing errors (ex. 4.02(8,95) no space).
- We thank the reviewer for the observation, the lines have been corrected, and adjustment of the font size was required to allow spacing.
Table 7, 8
Some lines are missing.
- We thank the reviewer for the observation, the lines have been corrected.
L375 L377 / spacing errors
- Thank you. the two spacing errors have been corrected.
L437 explanation (?)
- Thank you, the additional word has been deleted.
Round 2
Reviewer 2 Report
Comments and Suggestions for Authors
I am not questioning the authors' diagnosis of PPPD. I am pointing out that if PPPD patients have vestibular dysfunction, the SVV results and posturographic data would be affected not only by PPPD, but also by vestibular function. Since vestibular dysfunction in PPPD should not be an acute disorder during study participation, the absence of acute impairment is not a reason for the absence of vestibular dysfunction. This study lacks consideration from that perspective.
In the analysis of covariance between the SVV standard deviation while on-axis yaw rotation and the posturography measurements, which was significant only for sway length and VFY, the results of the analysis were not stated in the paper. There appears to be no data to suggest that it was not significant for other posturography measurements. There is also insufficient discussion as to why it was significant only for sway length and VFY.
The authors' explanation for estimating correlations for all two groups of normal and PPPD cases together is not convincing. The correlation between SVV accuracy and posturographic data, SVV accuracy and spatial anxiety, and SVV accuracy and DHI must clearly show that PPPD is different from normal adults.
Author Response
I am not questioning the authors' diagnosis of PPPD. I am pointing out that if PPPD patients have vestibular dysfunction, the SVV results and posturographic data would be affected not only by PPPD, but also by vestibular function. Since vestibular dysfunction in PPPD should not be an acute disorder during study participation, the absence of acute impairment is not a reason for the absence of vestibular dysfunction. This study lacks consideration from that perspective.
- All the patients had secondary PPPD, which imply vestibular dysfunction. The description of previous diagnosis is provided in the subsection 2.1 (lines 104 to 108).
- The evaluations were performed about 4 months after the symptomatic phase. We observed no nystagmus, with normal standard eye movement recordings, and symmetrical VOR responses to sinusoidal rotation in the dark (both at low and high frequencies) (Table 2), with adequate gain during visuo-vestibular stimulation and rotation with visual fixation. The findings on VEMPs are described in the manuscript.
- Since SVV is a test of vestibular function, we agree that posturography was affected by vestibular function, which is the main finding of the study. The vestibular function could be peripheral and or central. We state that further studies are desirable to ascertain the results.
In the analysis of covariance between the SVV standard deviation while on-axis yaw rotation and the posturography measurements, which was significant only for sway length and VFY, the results of the analysis were not stated in the paper. There appears to be no data to suggest that it was not significant for other posturography measurements. There is also insufficient discussion as to why it was significant only for sway length and VFY.
- The manuscript describes the findings, the descriptive statistics for the groups were as follows:
The authors' explanation for estimating correlations for all two groups of normal and PPPD cases together is not convincing. The correlation between SVV accuracy and posturographic data, SVV accuracy and spatial anxiety, and SVV accuracy and DHI must clearly show that PPPD is different from normal adults
- The statistical analysis was selected according to the design of the study.
- We did not hypothesize different correlations between the groups, but correlation between the variables on the full spectra of data. A second study should be designed to study specific differences as requested, including ad hoc power for analysis.
